# WHEN DOES REWARD DRIVE EXPLORATION?

## ABSTRACT

Traditional reinforcement learning (RL) methods encourage exploration by adding incentives such as randomization, uncertainty bonuses, or intrinsic rewards. Interestingly, meta-reinforcement learning (meta-RL) agents can develop exploratory behavior even when trained with a purely greedy objective. This raises the question: *under what conditions does greedy reward-seeking behavior lead to information-seeking behavior?* We hypothesize that three ingredients are essential: (1) **Recurring Environmental Structure**, where environments generate repeatable patterns that can be exploited if discovered; (2) **Agent Memory**, which enables past interactions to guide future performance; and (3) **Long-Horizon Credit Assignment**, which allows the delayed benefits of exploration to shape present decisions. Experiments in stochastic multi-armed bandits and temporally extended gridworlds demonstrate the need for recurrence, memory, and long-term credit. In short-horizon settings, however, exploration can arise from a *Pseudo-Thompson Sampling* effect, which mimics posterior sampling and obscures the role of temporal credit. In contrast, long-horizon environments reveal that explicit *Long-Horizon Credit Assignment* substantially improves returns. Our results suggest that structure, memory, and long horizons are critical for greedy training to induce exploration, highlighting these factors as key design considerations for effective meta-agents.

## 1 INTRODUCTION

Exploration is ubiquitous in humans and animals, yet unlike drives such as hunger or pain, there is no clear biological mechanism that directly incentivizes it. Neuroscience suggests that the brain balances exploration and exploitation through meta-learning, relying on repeated tasks, memory, and reward-driven adaptation Botvinick et al. (2018).

In reinforcement learning (RL), the exploration–exploitation trade-off is typically addressed by adding explicit incentives such as randomization ($\epsilon$-greedy), optimism (Upper Confidence Bound), or intrinsic rewards like curiosity-driven bonuses Auer et al. (2002); Pathak et al. (2019); Burda et al. (2018). These approaches treat exploration as a separate objective, distinct from exploitation.

By contrast, recent work in meta-reinforcement learning (meta-RL) shows that exploration can emerge even when agents are trained solely to maximize reward Duan et al. (2017); Wang et al. (2017). This raises a fundamental question: *under what conditions does pure exploitation give rise to exploratory behavior?*

We hypothesize that three factors are critical:

1. **Recurring Environmental Structure** — tasks repeat (or partially repeat), making early information useful later.

2. **Agent Memory** — policies can retain and exploit past interactions.

3. **Long-Horizon Credit Assignment** — learning connects information gathering to delayed payoffs.

However, apparent exploration under greedy training does not always originate from reward maximization itself. One confounding mechanism is what we term *pseudo-Thompson sampling*. In standard Thompson Sampling, an agent maintains a posterior distribution over rewards, samples from

it, and then acts greedily with respect to the sample—thus inducing exploration. Similarly, transformers trained via in-context learning can generate outputs that resemble samples from a pseudo-distribution conditioned on past interactions Hataya & Imaizumi (2024). When paired with a Deep Q-Network (DQN) Mnih et al. (2015) objective that selects the action with maximal predicted return, this process parallels Thompson Sampling, but does not arise from truly greedy motivations.

To disentangle these possibilities, we conduct controlled ablation studies in stochastic multi-armed bandits and temporally extended gridworlds. By systematically varying environmental recurrence, memory capacity, and cross-episode discounting, we test whether exploration emerges from reward-driven pathways or from artifacts of distributional modeling. Our results show that in short-horizon settings, pseudo-distributional effects such as pseudo-Thompson sampling dominate. In contrast, in long-horizon environments, reward-seeking behavior—when supported by recurrence, memory, and temporal credit—collects and exploits information without explicit exploration incentives. This mirrors mechanisms hypothesized in the human brain.

## 2 RELATED WORK

The exploration–exploitation trade-off is a longstanding challenge in reinforcement learning (RL) and sequential decision-making Sutton & Barto (2018); Lattimore & Szepesvári (2020); Thrun (1992). Early approaches relied on randomization techniques such as $\epsilon$-greedy or softmax action selection, as well as principled methods like Upper Confidence Bound (UCB) Auer et al. (2002) and Thompson Sampling (TS) Thompson (1933), which provide regret guarantees.

To extend exploration beyond the tabular setting, researchers introduced density modeling and pseudo-count methods Bellemare et al. (2016). Hashing-based exploration Tang et al. (2017) and successor representations Machado et al. (2020) have also shown strong performance in complex domains by encouraging novelty through intrinsic bonuses.

Intrinsic motivation mechanisms, such as curiosity-driven exploration, reward agents for visiting novel or unpredictable states Oudeyer & Kaplan (2007); Pathak et al. (2019). Random Network Distillation (RND) Burda et al. (2019) is a notable example, providing intrinsic bonuses in sparse-reward Atari games. More recent methods, such as SPIE Yu et al. (2023), combine state counts and trajectory structure to target exploration bottlenecks.

Meta-reinforcement learning (meta-RL) approaches exploration differently: rather than designing explicit bonuses, the goal is to learn strategies that adapt rapidly across related tasks Duan et al. (2017); Zintgraf et al. (2020). Notably, recurrent policies in meta-RL have been shown to explore effectively via reward maximization alone Duan et al. (2017); Wang et al. (2017). These works provide the foundation for our investigation.

A newer line of meta-RL research leverages transformers that attend directly to context rather than relying on recurrent policies Rentschler & Roberts (2025); Melo (2022). These models make it possible to exploit large-scale pretraining and to test how exploration depends on a finite context window.

In addition to introducing new empirical results, our work clarifies the boundaries of emergent exploration in meta-RL. We explicitly identify and test the conditions—recurring environmental structure, agent memory, and long-term credit assignment—under which greedy training produces exploration. To our knowledge, prior studies have observed emergent exploration but have not systematically articulated or validated these preconditions. Our findings show that when any of these factors is absent, greedily trained agents fail to explore effectively, underscoring their necessity for exploration to emerge.

## 3 BACKGROUND

### 3.1 REPEATED MARKOV DECISION PROCESSES (MDPs)

To study when exploration can emerge from pure exploitation, we adopt a repeated-task setting. A fixed, partially observable Markov decision process (POMDP)

$$M = (\mathcal{S}, \mathcal{A}, \mathcal{O}, P, \Omega, r, \gamma)$$

is sampled from a distribution, and the agent interacts with this *same* environment for multiple episodes. Here, $\mathcal{S}$ is the state space, $\mathcal{A}$ the action space, and $\mathcal{O}$ the set of observations available to the agent. The transition dynamics are given by $P$, the observation kernel by $\Omega$, the reward function by $r$, and $\gamma$ is the discount factor.

At the end of each episode, with probability $1/n$, a new parameterization of the environment is sampled, marking the start of a new task block. Thus, each block consists of a geometrically distributed number of episodes with mean $n$, during which the environment's parameters remain fixed. As a result, information gathered early in the block—such as the location of a goal or the identity of the best arm—remains valuable in later episodes. This regime enables agents to accumulate and exploit knowledge across episodes.

### 3.2 Meta-Reinforcement Learning (Meta-RL)

Traditional reinforcement learning (RL) trains agents to solve a single environment. In contrast, meta-reinforcement learning (Meta-RL) aims to produce agents that can rapidly adapt to new tasks. One approach, Model-Agnostic Meta-Learning (MAML), seeks an initialization that enables efficient weight updates at test time Finn et al. (2017). By contrast, $RL^2$-style methods avoid weight updates at test time. Instead, they employ recurrent neural networks (or transformers) to process the history of interactions and infer strategies for previously unseen environments, effectively "learning to learn" Duan et al. (2017); Wang et al. (2017). In this paper, we use the term meta-RL to refer to this second variant.

In the setting of repeated Markov Decision Processes (MDPs), the agent repeatedly interacts with the same environment. By leveraging experience across episodes, it can refine its policy for the current episode. This ability for cross-episode adaptation is key to understanding how seemingly greedy training can nonetheless foster exploratory behavior.

## 4 Methodology

Our experiments are designed to empirically test the hypothesis that exploration can emerge from pure exploitation objectives given certain preconditions. We choose short-horizon (Bandits) and long-horizon (Gridworlds) with repetitive runs in the same environment to train a transformer as the meta-RL agent with data derived from random policy acting in these environments. Training is done using the standard Deep Q-Networks (DQN) Mnih et al. (2015) algorithm.

### 4.1 Environments

We evaluate our hypothesis in two classes of environments: stochastic multi-armed bandits and temporally extended gridworlds. Both are framed within the repeated-task regime described in Section 3, where environment parameters remain fixed across multiple episodes within a block.

- **Multi-Armed Bandits:** Bandit environments are single-step decision problems, classically used to study the exploration–exploitation trade-off. We adopt the $K$-armed bandit setting, where each arm provides rewards from a stationary distribution (Bernoulli in our experiments). Each episode consists of a single step. Reward parameters remain fixed within a block. After each episode a new block begins with probability $1/n$. Thus, block lengths follow a geometric distribution with mean $n$. Larger $n$ yields greater recurrence—information from earlier episodes remains useful later—while smaller $n$ reduces recurrence. This design allows us to isolate the effect of environmental repetition on emergent exploration.

- **Gridworlds:** To study exploration in multi-step tasks, we use variants of the Frozen Lake environment. The agent begins in a start state and must reach a goal while avoiding holes, receiving reward $1/t$, where $t$ is the number of steps required to succeed. As in the bandit case, task blocks persist for a geometrically distributed number of episodes with mean $n$, after which a new map is sampled. Each map specifies random start, goal, and hole positions; trivial maps (goal reachable in fewer than three moves) and impossible maps (goal unreachable) are discarded. Grid sizes are sampled uniformly between $3\times3$ and $5\times5$, requiring agents to generalize across diverse maps without prior knowledge of structure.

Episodes are represented by stacking the sequence of actions, observations, and rewards. Episodes are then stacked into blocks, and blocks are further stacked until the transformer's context is filled (see Figure 1).

## 4.2 AGENT ARCHITECTURE

Agents use transformer-based value functions following Rentschler & Roberts (2025). The model ingests the most recent $X$ tokens (actions, observations, and rewards), which sets the effective memory capacity: smaller $X$ restricts memory to recent events, whereas larger $X$ enables cross-episode retention within a block. We initialize from the pretrained Llama 3.2 3B model and apply LoRA adapters ($r_{\text{LoRA}}=32$, $\alpha_{\text{LoRA}}=32$) for efficient fine-tuning.

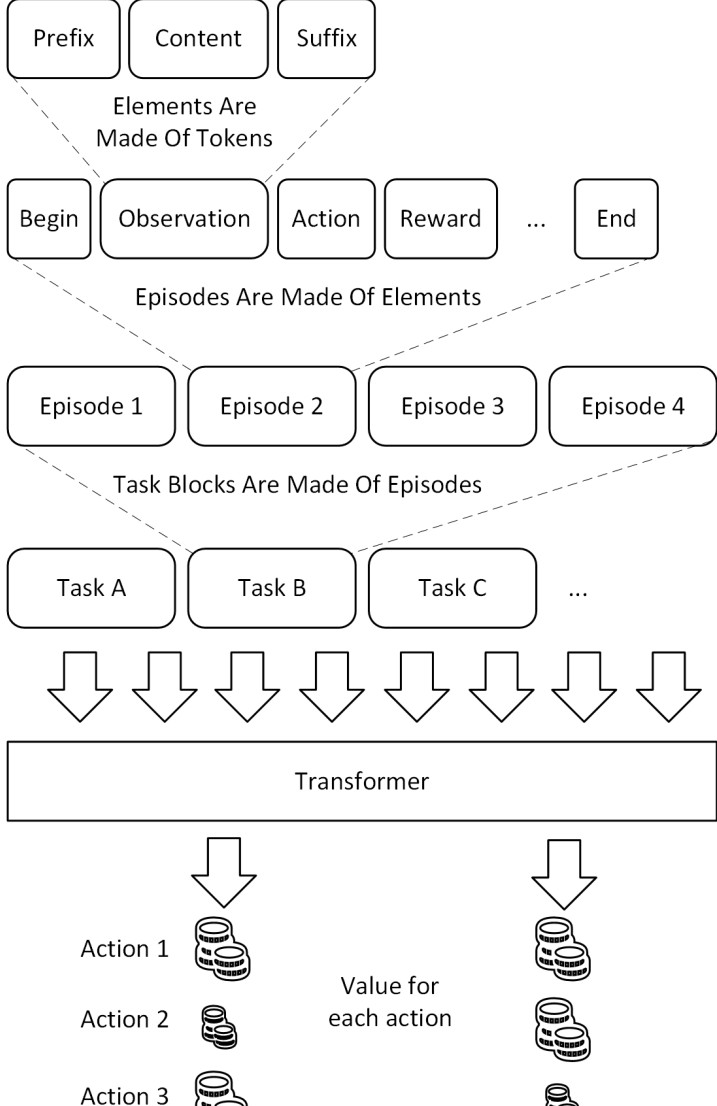

Figure 1: Setup for DQN meta-RL learning with a transformer. Tokens make up elements, which comprise episodes, which in turn make up task blocks. This is fed to the transformer which is trained to output the value of each action.

### 4.3 TRAINING

We train with DQN using a delayed target network for stability updated via Polyak averaging with factor $\tau{=}0.1$. The discounting scheme is tailored to the repeated-task (block) setting to control temporal credit assignment across episodes within a block:

- Within an episode, per-step discount is set to $1$.
- On the terminal transition of an episode, we apply $\gamma_{\text{episode}} \in [0, 1]$.
- On the terminal transition of a block boundary, we set the discount to $0$.

Thus, unlike the usual practice of setting the terminal-episode discount to $0$, we allow nonzero cross-episode credit *within* a block when $\gamma_{\text{episode}}{>}0$.

Unless otherwise noted, training runs for 400 iterations in bandits and 1200 iterations in gridworlds. Training is conducted offline: we generate $M$ data streams by running environments with a random policy until each stream contains at least $20X$ tokens. At each iteration, we sample a fixed number of streams to form a batch, keeping tokens-per-batch approximately constant across context sizes. We apply action masking so that only valid action tokens contribute to the DQN loss.

### 4.4 TESTING

Prior to evaluation, we seed the transformer's context with trajectories generated by a random policy in other environments. At test time, a new environment is sampled, and the model interacts for a fixed number of episodes. No additional exploration noise is introduced.

### 4.5 METRICS

Exploration is valuable only insofar as it translates into higher future reward. For this reason, our primary evaluation metric is the cumulative return achieved over a limited number of episodes. Policies that engage in effective exploration discover superior strategies and thereby accumulate greater total reward, whereas policies that fail to explore typically overfit to incidental experiences and converge to suboptimal returns. Measuring return directly therefore provides the most natural and principled way to assess the quality of exploration.

To complement this outcome-based perspective, we also analyze behavioral indicators of exploration. In the gridworld experiments, we visualize state visitation distributions. These heatmaps serve as heuristic evidence of exploration by revealing whether agents initially spread their visits broadly across the map before converging to efficient goal-directed paths. Such qualitative patterns allow us to compare emergent exploration to human-like strategies, highlighting whether agents exhibit the same exploratory-to-exploitative progression commonly observed in human learning.

### 4.6 BASELINES

Our aim is not to surpass conventional baselines in raw performance, but to show that emergent exploration through greedy exploitation crucially depends on recurrence, memory, and temporal horizon. Still, it is important to situate our results relative to established exploration strategies in order to calibrate their significance.

For the bandit environments, we therefore compare against canonical baselines: Thompson Sampling, $\epsilon$-greedy, a random policy, and an oracle with full knowledge of reward distributions. These comparisons allow us to position emergent exploration alongside both theoretically principled methods (Thompson Sampling) and widely used heuristics ($\epsilon$-greedy), while also bracketing performance between lower (random) and upper (oracle) bounds.

In the gridworld setting, there are no widely accepted meta-RL baselines that directly isolate exploration. A central advantage of meta-RL is its ability to generalize exploratory strategies to multi-step domains. Comparing against standard RL baselines would require online parameter updates during evaluation, introducing confounding factors such as update frequency, stability, and computational overhead—issues that are orthogonal to our study. To avoid these complications, we restrict our comparisons to two clear references: a random policy and an oracle policy.

## 5 EXPERIMENTS

We evaluate our hypothesis—that exploration can emerge from pure exploitation given the right structural conditions—using controlled ablation studies in multi-armed bandit and gridworld tasks. Across all experiments, we systematically vary: the degree of environment recurrence ($n$), agent memory capacity ($X$), and the temporal credit assignment horizon ($\gamma_{episode}$).

All results are averaged over multiple seeds: 10,000 runs for baselines and 1,000 for meta-RL in the bandit task, and 1,000 runs for baselines and 100 for meta-RL in gridworld. To demonstrate significance, we report 95% confidence intervals taken across all seeds.

To facilitate comparison and interpretation, we linearly normalize all reported rewards: a value of 1 denotes performance of an oracle agent, while 0 indicates the average performance of a random agent under identical conditions.

### 5.1 MULTI-ARMED BANDIT RESULTS

Figure 2 shows the performance of the meta-RL agent in a 3-arm bandit environment. In the 30 episode regime, the agent's cumulative returns are on par with the Thompson Sampling baseline.

**Effect of Recurring Structure** Table 1 shows that higher mean block lengths ($n$)—i.e., greater environment recurrence—enable agents to leverage early exploration for improved performance in later episodes. Performance drops sharply for smaller $n$, confirming that recurring structure is necessary for emergent exploration.

**Effect of Memory Capacity** Table 2 demonstrates that reducing the agent's memory capacity (context window $X$) leads to a sharp decline in cumulative reward. Below a critical threshold, exploration fails to emerge, confirming that sufficient memory is essential.

**Effect of Temporal Credit Assignment** When the temporal horizon is ablated ($\gamma_{episode} = 0$), meta-RL continues to display exploratory behavior, without appreciable reduction in performance.

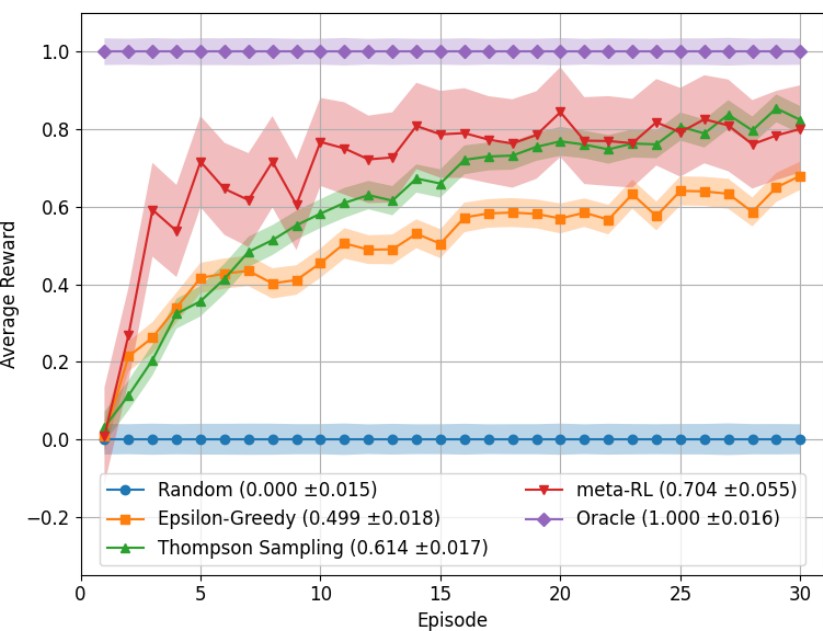

Figure 2: Reward per episode in the 3-arm bandit environment, comparing meta-RL ($n = 30$ $X = 1024$ $\gamma_{episode} = 0.9$) to baseline strategies. Shaded areas denote 95% confidence intervals. As Meta-RL interacts with the environment, its performance improves, demonstrating an ability to explore and then exploit.

| $n$ | $\epsilon$-Greedy | TS | Meta-RL |
|---|---|---|---|
| 30 | 0.499 ±0.018 | 0.614 ±0.017 | 0.704 ±0.055 |
| 10 | 0.354 ±0.020 | 0.339 ±0.019 | 0.509 ±0.064 |
| 3 | 0.147 ±0.027 | 0.092 ±0.025 | 0.325 ±0.082 |
| 1 | -0.083 ±0.041 | -0.066 ±0.041 | 0.043 ±0.130 |

Table 1: Cumulative reward as a function of mean block length $n$ in the 3-armed bandit environment.

| $X$ | $\epsilon$-Greedy | TS | Meta-RL |
|---|---|---|---|
| 1024 | | | 0.704 ±0.055 |
| 256 | | | 0.792 ±0.054 |
| 128 | 0.499 ±0.018 | 0.614 ±0.017 | 0.574 ±0.062 |
| 64 | | | 0.547 ±0.060 |
| 32 | | | -0.052 ±0.099 |

Table 2: Cumulative reward as a function of context window $X$ in the 3-armed bandit environment.

This lends credence to our theory that pseudo-Thompson Sampling is driving exploration. More discussion about this is available in section 6.

### 5.2 GRIDWORLD RESULTS

Figure 3 shows the performance of the meta-RL agent in the Frozen Lake gridworld. The agent learns exploratory policies improving from random toward the expert oracle.

**Effect of Recurring Structure** As with bandits, increasing block length $n$ in gridworlds yields higher cumulative reward (Table 3). The agent exploits repeated structure to improve over time.

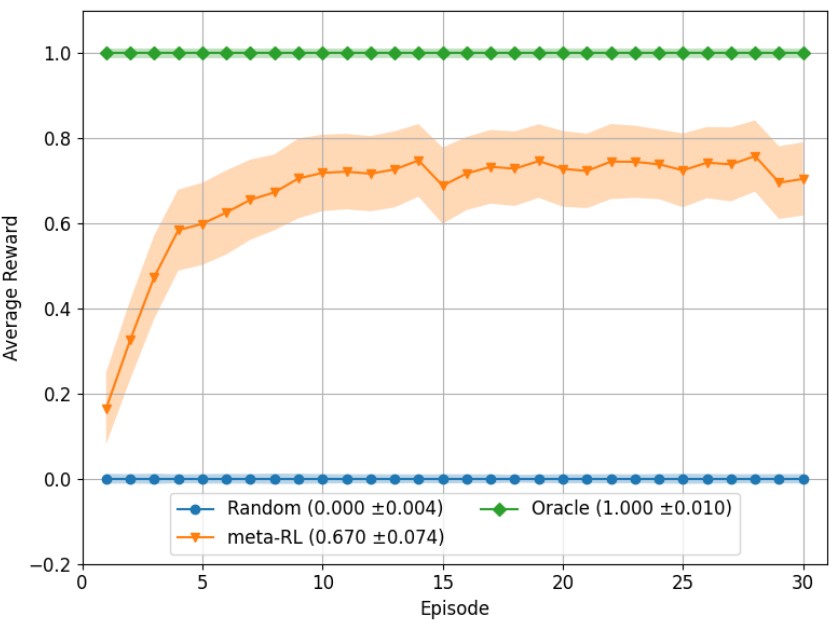

Figure 3: Reward per episode in the Frozen Lake gridworld, comparing meta-RL ($n = 30$ $X = 1024$ $\gamma_{episode} = 0.9$) to random and oracle strategies. Shaded areas denote 95% confidence intervals. The meta-RL agent achieves significant gains, demonstrating that it gathers information (exploration) that is later used to gather rewards (exploitation).

| $n$ | Meta-RL |
|---|---|
| 30 | 0.670 ±0.074 |
| 10 | 0.441 ±0.070 |
| 3 | 0.254 ±0.071 |
| 1 | -0.050 ±0.004 |

Table 3: Cumulative reward as a function of mean block length $n$ in Frozen Lake grid-worlds.

| $X$ | Meta-RL |
|---|---|
| 1024 | 0.670 ±0.074 |
| 256 | 0.120 ±0.056 |
| 128 | 0.047 ±0.026 |
| 64 | -0.001 ±0.033 |

Table 4: Cumulative reward as a function of memory window $X$ in Frozen Lake grid-worlds.

**Effect of Memory Capacity**  Reducing the context window $X$ also decreases agent performance in gridworlds, though the critical threshold for collapse occurs at higher $X$ than in bandits due to the increased episode length (Table 4).

**Effect of Temporal Credit Assignment**  Our results show that while meta-RL agents can exhibit exploratory behavior in bandit tasks even with $\gamma_{episode} = 0$, in gridworld environments, increasing the discount factor from zero to a nonzero value leads to a moderate improvement in overall performance (which is indicative of reward induced exploration), with the average reward rising from 0.408 ±0.089 to 0.670 ±0.074.

**State Visitation**  Figure 4 shows the state visitation distribution across block progression aggregated over 10 trials in the same environment with different randomly seeded contexts (percentages represent the relative number of time steps spent in a particular state). Early on, agents visit a broad range of states. As training progresses, visitation becomes concentrated along paths leading to the nearest goal, and agents appear to avoid hazardous states such as holes. Notably, exploration seems to expand outward from the starting position, gradually covering more distant regions.

## 6 DISCUSSION

Our findings provide strong evidence that exploration does not require explicit bonuses or injected randomness. Instead, it can emerge naturally from purely exploitative training when the right conditions are met. Specifically, when environments exhibit recurring patterns, agents possess sufficient memory, and learning propagates reward across a long enough horizon, greedy policies consistently engage in information-seeking behavior.

A central challenge is disentangling the role of **pseudo-Thompson Sampling** from that of **long-horizon credit assignment**. In bandit tasks, exploration persists even when cross-episode reward propagation is disabled—a signature of pseudo-TS rather than reward-driven exploration. In grid-

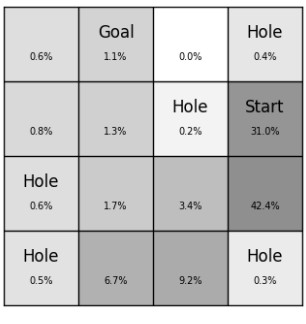
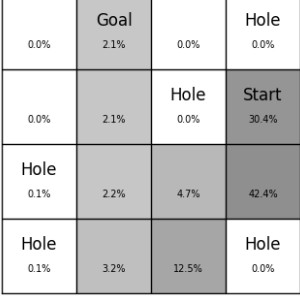

|        (a) Early        |        (b) Middle        |        (c) Late        |

Figure 4: State visitation heatmaps in the same gridworld (early episodes 1–3, middle 4–7, and late 8–30). Darker shading indicates more time spent in that state. Early episodes show exploration around the start; middle episode avoid holes and explore further out; late episodes follow discovered paths to goal.

worlds, however, pseudo-TS alone is insufficient: when $\gamma_{\text{episode}} > 0$, higher returns emerge, showing that long-horizon credit assignment becomes essential. Thus, pseudo-TS can drive exploration in short-horizon domains, but long-horizon tasks require temporally extended credit to support effective exploration.

Why is this the case? Pseudo-TS depends on accurate distributional estimates. In short episodes with relatively few action sequences, the value distribution remains tractable, enabling pseudo-TS to approximate posterior sampling effectively. In long episodes, however, the number of possible action sequences grows exponentially—millions in the case of gridworlds. This makes accurate distributional modeling intractable, leading to poor estimates, suboptimal exploration, and degraded performance. Long-horizon credit assignment is therefore critical for sustaining exploration in complex, temporally extended tasks.

## 7 LIMITATIONS

Our study focuses on environments with explicit recurring structure, sufficient agent memory, and long-term credit propagation. While we argue that these conditions are necessary, they may not be sufficient; exploration could still fail to emerge for other reasons. Also, further research is needed to evaluate the robustness and scalability of emergent exploration in more complex and diverse domains.

## 8 CONCLUSION

We identify and validate three necessary conditions for emergent exploration under greedy training: recurring structure, agent memory, and long-horizon credit assignment. Through controlled ablations in bandits and gridworlds, we show that removing any one of these factors eliminates reward-driven exploration. While generative models can induce a pseudo-Thompson Sampling effect that sustains exploration in short-horizon tasks, long-horizon tasks require temporally extended credit assignment to incentivize exploration. Our work clarifies when and why exploitation alone can produce information-seeking behavior, shifting the design focus from explicit bonuses to architectures and training regimes that leverage structure and memory.

We hope these findings encourage continued exploration (pun intended) of how context, memory, long horizons, and distributional modeling contribute to emergent exploratory behavior. Ultimately, this line of research brings us closer to agents that explore simply because it is the most effective way to maximize reward.

## REPRODUCIBILITY STATEMENT

We have made extensive efforts to ensure the reproducibility of our work. The environments used in our experiments—stochastic multi-armed bandits and gridworlds—are described in Section 4, where key design choices are explicitly documented. This includes the architecture and training methodology of our meta-RL agents, along with transformer configurations, context window sizes, discounting schemes, hyperparameter settings, and evaluation metrics. Section 5 details the ablations performed, the use of multiple trials, and the normalization of results. Additional resources, including complete training code, environment generation procedures, and implementation scripts, are provided in the supplementary materials. Finally, we present visualizations of agent learning progress (Figures 4 and 3) and analyses of ablation studies to further support replication and verification of our findings.

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

## A    USE OF AI

Large language models (LLMs) were used to assist in the preparation of this text. Specifically, LLMs helped check spelling and grammar and suggested alternative phrasing for certain sections. On occasion, AI tools were also used to identify supporting references. However, the core ideas and arguments presented in this work originated with the authors themselves.

