# OpenReview forum: "When Does Reward Drive Exploration?"
_ICLR.cc/2026/Conference — ICLR 2026 Conference Withdrawn Submission_

### Official Review · Reviewer_9PHu · 2025-10-29

**Soundness:** 2
**Presentation:** 2
**Contribution:** 2
**Rating:** 2
**Confidence:** 4

**Summary:**

This paper studies an interesting question: under what condition does an agent trained with purely greedy objective can exhibit exploratory behavior. The authors hypothesized three key ingredients: (a) recurrent environmental structure; (b) agent's memory capability; (c) long-horizon credit assignment. Having trained a transformer-based DQN, the authors identified the emergence of a "Pseudo-Thompson sampling" effect, leading to exploration. The horizon of credit assignment is a more flexible requirement, where short-horizon is sufficient for tasks do not require long-horizon credit assignment (e.g. bandits), but is necessary in long-horizon grid worlds.

**Strengths:**

- The problem under study is important and timely to the field, and the purely empirical investigation is a useful approach.

**Weaknesses:**

- The experimental setup (both bandit and grid world) is based on training an agent offline on a dataset generated by a random policy. A random policy provides full-support data coverage that an online agent, which must contend with its own exploration bias, would never have. The paper's claims are about emergent exploration in a learning agent, but the experiment is fully disconnected from the online learning process where exploration is actually a problem. It is highly likely the findings are an artifact of this data-coverage "oracle" and do not generalize to any practical online RL setting.
- Apart from sample efficiency of learning, there is no concrete quantitative evaluation of the exploration performance.
- The authors use a massive, pre-trained transformer model to solve trivial tasks like three-armed bandits and 5x5 grid worlds. I find it hard to justify the architectural choice, which greatly hinders interpretability and induces confounding factors.
- The key concept, "Pseudo-Thompson sampling", is vaguely described to begin with and left unsubstantiated. This concept is central to the paper's claims but is never analyzed or properly disentangled, leaving the core part of the paper's argument speculative.

**Questions:**

See above.

---

### Official Review · Reviewer_NEsn · 2025-11-01

**Soundness:** 2
**Presentation:** 2
**Contribution:** 1
**Rating:** 2
**Confidence:** 4

**Summary:**

This work aims to clarify under which conditions training an RL method to maximize the standard RL objective will lead to the RL algorithm automatically being explorative. This is motivated from previous works such as Duan et al. (2017) and Wang et al. (2017) in meta RL that aim to meta learn exploration strategies. The work in particular focuses on the characteristics of the environment structure (whether there is recurring structure), the agent structure and memory and the horizon of credit assignment. The work claims to show that these factors are necessary conditions for the emergence of exploration (but not necessarily sufficient as explained in the limitations). The considered a Q-learning based agent that has a transformer based Q-function. They performed experiments on a three-armed bandit task as well as on procedurally generated sequential grid worlds (the frozen lake gridworld with repeated structure with an introduced degree parameter n).
For training, they gather data using a random policy, store this data, and then train their policy offline on this data set. For testing, they sample a new environment, and run the transformer based model on this new environment (i.e., no new updates are performed).
Regarding the results, for example in the bandit tasks, they perform 30 episodes, and MetaRL learns initially faster than Thompson sampling and Epsilon greedy, but by the 30 episode mark MetaRL seems to have plateaued and Thompson sampling caught up (and appeared to still have a rising performance). On the gridworld tasks, they compare against a random policy and an oracle policy that gets perfect rewards, but do not compare against other algorithms (their method achieves around 70% of the oracle policy).

**Strengths:**

- The topic is interesting.

**Weaknesses:**

- The results did not seem substantial to me as only simple bandits and gridworlds are considered. The page limit is not filled.

- I was not convinced by the experimental setup that first gathers data using a random policy and then does all training offline. A key component of exploration is the data gathering, but in the current setting this is done with a fixed random policy.

- Some claims were not convincing to me. The work mentions that they claimed to have found necessary conditions; however, the experimental results only pertain to the setup considered in this paper (Q-learning, a particular type of transformer value model, grid worlds, etc.). If the setup is changed, the results may be different; therefore it is not possible to say from the current results that the conditions are truly necessary. Moreover, (while this is mentioned in the limitations) it would be better to have sufficient conditions, so that researchers could know how to induce exploration. The current results do not necessarily tell us how to achieve emergent exploration; they only claim three things that the paper claims are useful to include (structure in the environment, memory, and long horizons). And these three things are quite basic in the sense that RL works may typically include memory and long horizons anyhow, and the environment is not something that is changed from the algorithmic perspective, so the usefulness of the current claims is not fully clear to me.

- The result in Figure 2 seemed inconsistent to me (perhaps it’s my misunderstanding?). The MetaRL version rises faster in the beginning, but what we would expect instead is that as learning progress, it becomes better at exploration, and then outperforms the existing methods of Thompson sampling and epsilon greedy. Thus it is not convincing to me. Also, the x-axis is stopped right as Thompson sampling catches up to MetaRL, indicating that if the number of episodes is increased, MetaRL may stay stuck at it’s current performance while Thompson sampling may continue improving and outperform MetaRL due to better exploration. **Can you run it for more episodes (around 60 episodes or so)?**

- The results in Figure 4 seem strange to me. The heatmap indicates the amount of time spent in each state. For example, in (c), it says it spent 30% of the time at the start state, and 40% of the time at the state right next to it. But there are many states remaining until it reaches the goal state. **Why would it spend ~70% of the time moving back and forth between the start state and the state next to it?**

**Questions:**

See the weaknesses.

---

### Official Review · Reviewer_BfTA · 2025-11-01

**Soundness:** 2
**Presentation:** 2
**Contribution:** 1
**Rating:** 2
**Confidence:** 3

**Summary:**

The paper empirically studies the emergence of exploratory behavior in policies when they are trained to perform meta-RL. Three criteria are cited and studied to determine whether such behavior will appear.

**Strengths:**

The problematic at hand is interesting, and the paper is overall clearly written.

**Weaknesses:**

Here are a number of weaknesses that I have identified. Some of these may be due to misunderstandings and should therefore be considered as questions.
1. The paper refers to the exploratory (information-seeking) behavior of policies throughout the paper. However, this behavior is never formally characterized. It is therefore difficult to validate the empirical results.
2. I feel that two types of exploration are confused in the paper. First, exploration in the sense of MDPs, which can be measured by the variety of states visited and observed. We know that it is necessary to visit these states in order to identify the optimal policy as an intermediate stage for learning (the optimal policy does not explore). Second, exploration in the sense of POMDPs/Meta-RL. Here, an optimal policy will sometimes have to perform actions that can be described as exploratory in order to discover the hidden state and eventually perform actions that yield rewards. Here, it is the optimal behavior and not a learning step. I believe this distinction is very important in the context of Meta-RL, but it is not made.
3. The algorithm description is very limitted for non-expert in meta-RL.
4. I don't think the experimental method allows us to evaluate the exploratory behavior of policies. It is mainly an ablation study on the influence of credit assignment (gamma) and policy memory (number of past observations considered) on the expected return. The results are also fairly trivial.

**Questions:**

Weaknesses can be considered questions.

1. Can the authors clarify their definition of exploration mathematically? What is a pure-exploration policy? What is its expected return?

2. Can the authors clarify this statement :

> Exploration is valuable only insofar as it translates into higher future reward. For this reason, our primary evaluation metric is the cumulative return achieved over a limited number of episodes. Policies that engage in effective exploration discover superior strategies and thereby accumulate greater total reward, whereas policies that fail to explore typically overfit to incidental experiences and converge to suboptimal returns. Measuring return directly therefore provides the most natural and principled way to assess the quality of exploration.

I agree that exploring eventually leads to higher return but this neglects the fact that while exploring a policy leads to lower expected return. The point of the exploration-exploitation is that the expected return is balanced against the entropy of states (or other exploration measure). It means that exploration cannot be measured with the expected return.

3. Can the authors elaborate on the evolution of the distribution of states (uniform or not) to quantify the resulting exploration/information-seeking behavior of policies?

4. Are results expected to generalize beyond the particular algorithm and architecture?

---

### Official Review · Reviewer_6fQj · 2025-11-02

**Soundness:** 1
**Presentation:** 1
**Contribution:** 1
**Rating:** 2
**Confidence:** 2

**Summary:**

The paper investigates the conditions under which a meta-RL agent, trained with a purely greedy (exploitation) objective, can learn to explore. The authors hypothesize three necessary factors: recurring structure, agent memory, and long-horizon credit assignment. They test this in bandits and gridworlds, finding that in short-horizon tasks exploration emerges from a "pseudo-Thompson Sampling" effect, while in long-horizon task, explicit long-term credit assignment is required.

**Strengths:**

The paper asks a clear and fundamental question about emergent exploration in meta-RL. Using two distinct environments to isolate short-horizon vs. long-horizon effects is a good experimental design choice. The identification of a pseudo-Thompson Sampling effect, distinct from reward-driven credit assignment, is an interesting conceptual contribution.

**Weaknesses:**

The paper's central claim is fundamentally undermined by its choice of experiments, which is insufficient to support any conclusions about emergent exploration. I think the main weakness is how exploratory behavior is tested, the environments are overly simplistic and small.
In such trivial environments, a random policy provides near-complete coverage of the small state-action space. The agent is not necessarily learning to explore in a reward-driven manner, but it might simply perform in-context learning on an offline dataset that already contains all the necessary information. A large model (especially a pretrained 3B Llama) can easily extrapolate the optimal policy from these random trajectories.
Achieving good performance in these benchmarks is therefore not necessarily sign of emergent exploratory behavior.

**Questions:**

- The ablation tables (e.g., Table 1, Table 2) are missing default values. For example, when ablating the block length $n$ in Table 1, what are the default context window $X$ and $\gamma_{episode}$ used for those runs?
- How are the $\epsilon$-greedy and Thompson Sampling baselines implemented? Are they standard tabular algorithms, or do they also use the same transformer architecture as the meta-RL agent? This is critical for understanding if the comparison in Figure 2 is fair.
- Could a simple offline RL algorithm (like basic Q-learning on the random data) or even a behavioral cloning baseline achieve similar results? This would help clarify if the meta-learning aspect is doing any heavy lifting.
- Given that the environments are simple enough to be well covered by a random policy, how can you be certain the agent is learning to explore rather than just extrapolating the optimal policy from the random data?

---

### Official Review · Reviewer_wtwi · 2025-11-03

**Soundness:** 2
**Presentation:** 1
**Contribution:** 1
**Rating:** 0
**Confidence:** 3

**Summary:**

This paper explores the RL conditions in which the greedy reward-seeking behavior lead to information-seeking behavior. The authors utilized the DQN approach to train a Llama 3.2 3B model (with LoRA) to conduct their experiments on multi-armed bandits and gridworlds settings.

**Strengths:**

1. The paper presented and attempted to answer an interesting question: **Under what conditions does greedy reward-seeking behavior lead to information-seeking behavior?**

**Weaknesses:**

1. The experiments are confined to stochastic multi-armed bandits and simple gridworlds. While these are classic environments for studying exploration, they are **toy problems** and are not very complex.


2. The agent architecture is a pretrained Llama 3.2 3B model that is then fine-tuned. This introduces a major unexamined variable. It is unclear how much of the agent's performance, especially the **pseudo-Thompson sampling** effect, is an emergent property of the meta-RL training versus a capability already present in the massive pretrained model.

3. Overall, it remains unclear about the mechanisms behind pseudo-Thompson sampling effect due to **limited experimentation**. Moreover, the authors did not introduce the concept nor contrast it well enough, mathematically.

**Questions:**

1. What were the reasons behind the number of fine-tune iterations? What if you decrease them to 100s? Would the trend still hold? It is dificult to tell if the effects, you delineated, are true when you are experimenting with a large model.

2. Could you please contrast what you mean by **pseudo-Thompson sampling** with Thompson sampling mathematically?

---

### Official Review · Reviewer_jFd4 · 2025-11-04

**Soundness:** 2
**Presentation:** 2
**Contribution:** 2
**Rating:** 2
**Confidence:** 4

**Summary:**

This paper investigates when and why meta-reinforcement learning (RL) agents develop exploratory behavior through purely greedy training objectives without explicit exploration incentives. The authors hypothesize that three conditions are necessary for emergent exploration: recurring environmental structure (where tasks repeat, making early information valuable later), agent memory (enabling retention of past interactions), and long-horizon credit assignment (connecting information gathering to delayed rewards). The paper introduces the concept of pseudo-Thompson sampling, a confounding mechanism where transformers trained with in-context learning generate outputs resembling samples from a pseudo-distribution, which can induce exploration-like behavior independent of reward-driven mechanisms. Through controlled experiments in multi-armed bandits and Frozen Lake gridworlds, the authors systematically ablate environmental recurrence, memory capacity, and cross-episode discounting. Results demonstrate that all three hypothesized factors are necessary, with pseudo-Thompson sampling dominating short-horizon settings while long-horizon credit assignment becomes essential in temporally extended tasks.

**Strengths:**

Originality

- The core hypothesis is well-articulated and theoretically grounded, explicitly identifying three necessary conditions that prior work has observed but not systematically validated.
- The introduction of pseudo-Thompson sampling as a confounding mechanism is conceptually insightful.

Quality

- The experimental design is reasonable, employing controlled ablation studies that systematically vary environmental recurrence, memory capacity, and temporal credit assignment across two toy domains.
- The use of normalized rewards facilitates clear interpretation, and reporting 95% confidence intervals with large sample sizes enhances statistical reliability.

Clarity

- The paper is easy to follow. However, the presentation could be more precise (see weaknesses).

Significance

- The distinction between short-horizon and long-horizon exploration mechanisms represents an interesting finding that clarifies when different mechanisms dominate.

**Weaknesses:**

1. The pseudo-Thompson sampling mechanism, while conceptually interesting, lacks rigorous theoretical justification. It is introduced informally and never precisely defined, making it difficult to distinguish from standard uncertainty-based exploration.

2. The claim that transformers generate outputs resembling samples from a pseudo-distribution is not entirely convincing, and the paper provides no direct empirical evidence demonstrating this mechanism in their trained agents. The authors do not measure or visualize value distributions to confirm that transformers are actually performing distributional modeling, nor do they analyze attention patterns or internal representations to support the pseudo-sampling hypothesis. The distinction between pseudo-Thompson sampling and reward-driven exploration relies primarily on the presence or absence of cross-episode credit, but it is possible that other mechanisms could explain similar patterns.

3. The experimental scope presented in the paper is narrow. Both environments are relatively simple with discrete action spaces and short episodes, raising questions about generalization to continuous control and/or high-dimensional observation spaces.

3. The paper lacks comparisons with other meta-RL approaches, intrinsic motivation methods adapted for meta-RL, or established benchmarks like Meta-World or Atari, limiting assessment of practical significance.

4. Implementation and methodological details presented in Section 5 are missing or unclear. The paper does not report key hyperparameters, including learning rates, batch sizes, number of data streams, or target network update frequencies.

5. The offline training procedure generates data from random policies, but it remains unclear whether this choice affects which exploration mechanisms emerge compared to online training.

6. The choice to use Llama 3.2 3B with LoRA fine-tuning is not justified, and there is no analysis of how model size or pretraining affects emergent exploration.

7. The theoretical understanding of the study presented in the paper is limited. The paper demonstrates empirically that the three conditions are necessary but provides no formal proofs or theoretical guarantees. The conditions may not be sufficient, as the authors acknowledge, but there is no discussion of what additional factors might be required. The explanation for why pseudo-Thompson sampling fails in long-horizon settings invokes the intractability of distributional modeling over exponentially many action sequences, but this is presented as speculation rather than demonstrated analysis. The paper lacks theoretical predictions about critical thresholds for memory capacity or block length, making the findings primarily descriptive.

**Questions:**

1. Can the authors provide direct empirical evidence that their trained agents perform pseudo-Thompson sampling? For instance, can you visualize the value distributions predicted by the transformer and show that they exhibit meaningful variance that drives exploration?

2. The authors mention in the limitation section that the three identified conditions might not be sufficient for emergent exploration. Are there scenarios where all three conditions are met but exploration still fails to emerge?

3. How do model size and pretraining affect emergent exploration? Have the authors tested smaller models or models without pretraining to isolate the role of scale and initialization?

4. How easy/difficult is it to extend the analysis to more complex environments, including continuous control tasks, high-dimensional observations, or longer horizons to assess generalization of your findings?

5. How do your results compare to other meta-RL methods, intrinsic motivation approaches, or established benchmarks? This would help position the practical significance of emergent exploration relative to explicit exploration bonuses.

---

### Author Response · Authors · 2025-11-13
**Thank you for reviews**

We sincerely thank the reviewers for their thoughtful and constructive feedback. Your comments have been invaluable in pointing us in the right direction for improving our work.

Several of you noted that our description of pseudo-Thompson Sampling (pseudo-TS) was not sufficiently well defined or validated. What may not have been clear is that pseudo-TS arose as an observed effect rather than from a formal theoretical motivation. We agree that it would be helpful to add experiments showing the action distribution and how it concentrates after experience, and we plan to do so in future revisions.

Relatedly, you asked how we can validate that short-episode exploration is driven by pseudo-TS and long-episode exploration is driven by reward-based exploration. We agree that this requires additional analysis and experimentation, and we will need to think more carefully about how to design diagnostics for this distinction. We would be very interested in any suggestions you might have on this point.

We also acknowledge the concern that our experimental environments are simple. While this is true, the parameterization space is very large: we allow many different grid sizes with different starting points, end points, and hole positions. There is no way that all parameterizations are present in the random contexts used for training. Thus, the transformer must learn “how to solve general grid worlds” (yes this is in-context learning that is really the point) rather than simply memorizing specific parameterizations. We recognize that this motivation should have been explained more clearly and that extending to larger or continuous environments would further strengthen the work.

Several reviewers asked for comparisons with additional algorithms and baselines. To our knowledge, there are few, if any, directly comparable studies of meta-RL exploration that would serve as ideal baselines. However, we agree that traditional RL methods with explicit exploration (e.g., ε-greedy or other exploration strategies) could be used as comparison points, especially if we carefully examine their sample efficiency when run on new parameterizations of the environment. We will consider such comparisons in future versions and would welcome suggestions on baselines you deem most relevant.

Finally, you requested clearer implementation details for Thompson Sampling, ε-greedy methods, and the relationship between our approach and DQN-style value propagation across episodes (as in RL²). We can certainly add more detail. We implement a standard DQN algorithm with cross-episode value propagation and contextual history (that is it!); as in RL², agents trained in this way are effectively meta-agents. We see that this connection may not have been sufficiently explicit in the current draft.

Thank you again for the time and care you invested in reviewing our work. Because we are unable to fully address these concerns within the allotted time, we have decided to withdraw the paper for now. We remain very grateful for your feedback, and we will carefully consider any further ideas you may wish to share as we continue to develop this line of research.

---

### Note · Authors · 2025-11-24

I have read and agree with the venue's withdrawal policy on behalf of myself and my co-authors.